behaviour/cognition

agent-based model, ephemeral resources, predictability, search behaviour, synchrony, temporal availability

**Authors for correspondence:**
Benjamin Robira
e-mail: Benjamin.robira@normalesup.org
Simon Benhamou
e-mail: simon.benhamou@cefe.cnrs.fr

[†]Associated to Cogitamus lab.
[‡]Contributed equally and are listed in alphabetical order.

# Foraging efficiency in temporally predictable environments: is a long-term temporal memory really advantageous?

Benjamin Robira[1], Simon Benhamou[1,†], Shelly Masi[2], Violaine Llaurens[3,‡] and Louise Riotte-Lambert[4,‡]

[1]Centre d'Ecologie Fonctionnelle et Evolutive, Université de Montpellier and CNRS, Montpellier, France
[2]Eco-anthropologie, Muséum National d'Histoire Naturelle, CNRS, Université de Paris, Paris, France
[3]Institut de Systématique, Evolution, Biodiversité, CNRS-École Pratique des Hautes Études, Muséum National d'Histoire Naturelle, Université Pierre-et-Marie-Curie, Paris, France
[4]Institute of Biodiversity, Animal Health and Comparative Medicine, University of Glasgow, Glasgow, UK

BR, 0000-0002-3168-6573; SB, 0000-0003-0803-8559; SM, 0000-0002-5692-0645; VL, 0000-0003-1962-7391; LR-L, 0000-0002-6715-9898

Cognitive abilities enabling animals that feed on ephemeral but yearly renewable resources to infer *when* resources are available may have been favoured by natural selection, but the magnitude of the benefits brought by these abilities remains poorly known. Using computer simulations, we compared the efficiencies of three main types of foragers with different abilities to process temporal information, in spatially and/or temporally homogeneous or heterogeneous environments. One was endowed with a *sampling* memory, which stores recent experience about the availability of the different food types. The other two were endowed with a *chronological* or *associative* memory, which stores long-term temporal information about absolute times of these availabilities or delays between them, respectively. To determine the range of possible efficiencies, we also simulated a forager without temporal cognition but which simply targeted the closest and possibly empty food sources, and a perfectly prescient forager, able to know at any time which food source was effectively providing food. The *sampling*, *associative* and *chronological* foragers were far more efficient than the forager without temporal cognition in temporally predictable environments, and interestingly, their efficiencies increased with the level of temporal heterogeneity. The use of a long-term temporal memory results in a foraging

efficiency up to 1.16 times better (*chronological* memory) or 1.14 times worse (*associative* memory) than the use of a simple *sampling* memory. Our results thus show that, for everyday foraging, a long-term temporal memory did not provide a clear benefit over a simple short-term memory that keeps track of the current resource availability. Long-term temporal memories may therefore have emerged in contexts where short-term temporal cognition is useless, i.e. when the anticipation of future environmental changes is strongly needed.

## 1. Introduction

As the local density of resources changes both in time and space, numerous animal species have evolved abilities to process and memorize information towards reducing uncertainties about the spatio-temporal resource distribution [1,2]. A forager with such abilities is not limited to just resolve when it should leave a profitable place [3,4] to maximize its efficiency. It can take advantage of its previous experience to infer *where* (at which location), *what* (which type of resource) and *when* (at which time) resources occur. An ample body of literature (reviewed in [5]) has focused on the benefits of a spatial memory (which stores *where* information), in association with an attribute memory (which stores *what* information, e.g. type of food source, water hole and shelter). A number of models have specified the contexts in which a spatial/attribute memory substantially improves foraging success (e.g. when resources are scarce and patchy or when the sensory field is limited; [6]). In models of spatial memory, the temporal component has often been ignored or restricted to basic assumptions, e.g. animals associate two events in memory when they occur simultaneously [7], or the memory of an event simply decays with time [8]. Yet, animals may demonstrate far more advanced temporal cognition [9,10]. For example, it has been shown experimentally that animals could display an episodic-like memory [11–13], even if the exact ways temporal information can be stored in memory remains debated [14–17]. The contribution of a temporal memory (which stores *when* information) to foraging efficiency has been only marginally investigated to date [18]. The comparative benefits that a forager may obtain by relying on different types of temporal memory thus remain poorly known. This dramatically limits our understanding of the selection pressure that acts on temporal cognition.

The benefits brought by a temporal memory are likely to depend on the predictability of the different resources [19]. When resources are always available (e.g. shelter or permanent surface water [20]), a temporal memory is unlikely to be useful. When resources continuously regrow (e.g. grass patches [21,22]), a forager would benefit from only keeping track of recently depleted places to revisit them only after a delay [23,24]. By contrast, if resources are ephemeral but renew in some predictable way (e.g. seasonally), the foraging efficiency should be strongly modulated by the type of temporal memory used to infer when resources would be available. While seasonal variation in resources is striking in temperate, arid or cold areas, it also occurs in tropical forests [25]. For example, edible fruits of tropical trees are generally available only for a few days to weeks each year, which challenges frugivorous animals [26,27]. Thus, as temporal periodicity in resources availability occurs in many environments, selection pressure on temporal memory is likely to be widespread in many active foragers.

When exploiting seasonal resources, both a long-term (or reference) and a short-term (or working) temporal memory can be advantageous. A long-term temporal memory stores information on how the timing of production of a given type of resource correlates either with abiotic temporal proxies such as the temperature or photoperiod [28] (*chronological* memory) or with biotic temporal proxies [19,29], including long-term contingencies between resource type availabilities (i.e. the presence of one resource gives an immediate or future clue on the presence of another resource; *associative* memory). A short-term temporal memory stores information on resource availability based on the forager's experience of recently visited places (*sampling* memory). An animal endowed with a *chronological*, *associative* or using *sampling* memory can benefit from the fact that a periodically renewing type of resource is not available at random in time but occurs at specific absolute or relative times, or through serially correlated time series, respectively. For example, people may start looking for mushrooms because the weather becomes cold and humid (*chronological* knowledge) and/or because some time ago they have seen flowers of a species known to bloom some time before mushrooms appear (*associative* knowledge) and/or because they already encountered mushrooms at different locations they recently visited for other reasons (*sampling* knowledge).

We used computer simulations to compare the foraging efficiencies of five types of forager while they look for resources that are ephemeral but yearly renewed in a more or less predictable way: (i) a *basic*

forager, without any temporal knowledge of resource availability, (ii) a *sampling* forager, which uses a short-term memory of the resource types it fed on recently, (iii) an *associative* forager, which uses a long-term memory that stores the delays between the production times of different resource types; (iv) a *chronological* forager, which knows when any resource type is available by using a long-term memory that stores the actual times of resource production, and (v) a *prescient* forager, which always knows when resources are available at any location. The *basic* and *prescient* were considered only to assess the theoretical minimum and maximum efficiencies that can be reached in a given context.

# 2. Material and methods

We modelled the behaviour of a single forager and estimated its foraging efficiency in various types of environments. The model was implemented in C++ using the Code::Blocks v. 17.12 interface. Food sources may correspond to any ephemeral, periodically renewing resource. To add environmental context, we considered here that food sources were fruit trees of different tree species. Furthermore, to avoid interferences that may blur the differences of benefits brought by different types of temporal memory, we deliberately ignored or oversimplified any feature that was not directly related to the ability of a forager to guess when a given type of resource was available. In particular, as the contributions of spatial/attribute memory and temporal memory to foraging efficiency were likely to be independent of each other (the *sampling*, *associative* and *chronological* memories stored information about the phenologies of fruit tree species rather than individual trees, whereas the spatial/attribute memory stored location coordinates of individual tree locations), we considered for simplicity, but without the loss of generality, that animals were endowed with an accurate spatial/attribute memory, i.e. they perfectly knew the locations of the potential food sources and the type of resource provided at each location, but differed by their abilities to infer when resource was available.

## 2.1. Environment

The environment was defined as a $1000 \times 1000$ arbitrary length unit (lu) square. A hundred fruit trees for each of 30 tree species were distributed in space, corresponding to an overall density $\rho = 0.003$ tree per lu. Each tree species produced fruit only for a limited period, with specific starting and ending times, once for a 360 time unit (tu) period, corresponding to a 'year'. For simplicity, we considered that:

(1) Every tree yielded the same amount of fruit, whatever its species.
(2) The date of the fruit production peak for a species $k$, $M_k$, was randomly drawn in a uniform distribution over a year and was constant across years. Consequently (i) the time intervals between the production peaks of the different species were also constant across years and (ii) the production of a given fruit species was seasonal but the overall fruit production was not periodic (i.e. fruit were available all year long, but the fruit available at a given time of the year depended on the species).
(3) The fruit production by an individual tree $i$ belonging to species $k$ at location (in two-dimensional space) $\mathbf{x}_{k,i} = (x_{k,i}, y_{k,i})$ started at time $t(\mathbf{x}_{k,i})$. This time was drawn at random from an isosceles triangular distribution centred on $M_k$ with base length $d_k$: $t(\mathbf{x}_{k,i}) = M_k + d_k(U_i + U'_i - 1)/2$, where $U_i$ and $U'_i$ were two independent random variables drawn from a uniform distribution between 0 and 1. The base length, $d_k$, corresponds to the temporal dispersion of fruit production by species $k$. It is hereafter referred to as the 'desynchrony' level for this species, as a very low or very high $d_k$ value characterizes species for which trees produce fruit over a single month or over most of the year, respectively. We used an isosceles triangular distribution as it is an easy way to model the symmetry of fruit production around a peak that is often observed in the wild [25].
(4) Fruit at a given location $\mathbf{x}_{k,i}$ all appeared instantaneously at the same time $t(\mathbf{x}_{k,i})$ and, if not eaten, disappeared instantaneously 30 tu later, corresponding to a 'month'.

At the spatial level, we simulated either homogeneous or heterogeneous environments. In spatially homogeneous environments, the 3000 trees were randomly distributed over space, irrespective of their species, resulting in a Poisson distribution of tree density, both overall and for any species. In spatially heterogeneous (i.e. patchy) environments, the 100 trees of the same species were aggregated in 10 clusters. The cluster centres were randomly distributed over space, and the trees of a given cluster were distributed around its centre according to a circular bivariate Gaussian distribution with a

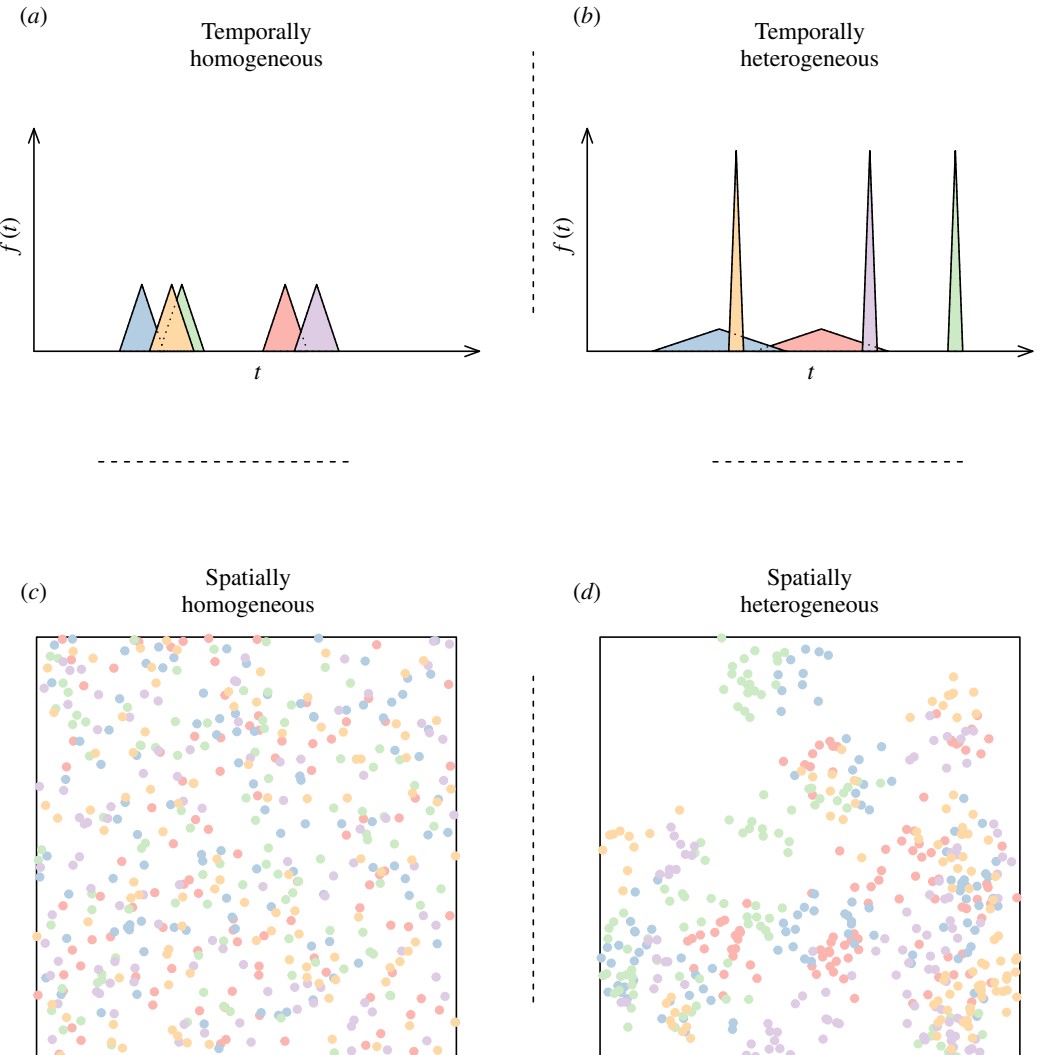

**Figure 1.** Spatially and temporally homogeneous and heterogeneous environments. To increase the readability, only five tree species are represented using different colours, whereas 30 species were considered in the simulations. The (a,b) panels show the starting dates of availability of fruit yielded by the different tree species over time. In a temporally homogeneous environment (a), all species had the same level of intra-specific desynchrony and were therefore characterized by the same triangular distributions. In a temporally heterogeneous environment (b), tree species with high (narrow and tall triangles) and low (wide and small triangles) intra-specific desynchrony levels were mixed (as a triangle represents a probability density distribution, its area is always equal to 1, whatever its shape). The (c,d) panels show the spatial distribution of fruit trees within a $1000 \times 1000$ lu environment. In a spatially homogeneous environment (c), the tree locations were uniformly distributed over space. In a spatially heterogeneous environment, trees belonging to a given species were aggregated by 10 to form clusters, which were uniformly distributed over space.

standard deviation of 20 lu. Both spatially homogeneous and heterogeneous environments could be either temporally homogeneous or heterogeneous (figure 1) and characterized by different levels of desynchrony. In temporally homogeneous environments, all tree species had the same desynchrony level, i.e. all species were characterized by the same $d_k$ value in the range of 36, 54, …, 324 tu (17 values ranging from $1/10$ to $9/10$ of a year by 0.05 step). In temporally heterogeneous environments, some of the tree species were characterized by a very low desynchrony level ($d_k = 36$ tu), while the others were characterized by a very high desynchrony level ($d_k = 324$ tu). The proportion $\alpha$ of tree species belonging to the first part was set to a value in the range of $1/16$, $2/16$, …, $15/16$, so that the mean desynchrony level, $\bar{d}_k = 36\alpha + 324(1-\alpha)$ tu, ranged between 54 and 306 tu. The level of temporal heterogeneity, $\alpha(1-\alpha)$, took its maximum value for $\alpha = 0.5$ ($\bar{d}_k = 180$), whereas the two most extreme cases ($\alpha = 0$, $d_k = 324$ tu and $\alpha = 1$ $d_k = 36$ tu), corresponded to temporally homogeneous environments. In each of the $2 \times (17 + 15) = 64$ types of environments we considered, we assessed the foraging efficiency of the

various types of foragers (see below) by running 300 batches of simulations. The location, $\mathbf{x}_{k,i}$, and production timing, $t(\mathbf{x}_{k,i})$, of each of the 3000 trees were redrawn for each simulation.

## 2.2. Foraging rules and efficiency

### 2.2.1. General rules

The assumptions driving the foraging behaviour for different types of temporal memory are illustrated in figure 2. A single forager moved straight from one tree to another located beyond its perceptual range. We set this range to a low value, $r = (2\sqrt{\rho})^{-1} \approx 9.13$ lu (mean tree density $\rho = 0.003$), corresponding to the mean distance to the closest fruit tree, because having a long-distance perception obviously decreases the benefits associated with memory [6]. The choice of the targeted tree was memory-dependent (see below). When a targeted tree yielding fruit was reached, the forager instantaneously emptied it. The trees recently visited (for the previous 15 tu) and those within the perceptual range, which were exploited en route, were never targeted. The moving speed was set to 150 lu tu$^{-1}$—a value slightly smaller than the threshold speed involving a systematic resource shortage. However, a temporary shortage may sometimes have occurred when the environment was submitted to high fluctuations of overall fruit abundance. In this case, the forager was momentarily kept at rest, mimicking the torpor period observed in some animals when their home ranges are fully depleted [30].

### 2.2.2. Choice of the targeted location (beyond perceptual range and not recently visited)

The *prescient* forager, which always knew the true status of every fruit tree, systematically targeted the closest productive tree. By contrast, the *basic* forager, which was unable to guess which species was in its fruiting phase, simply targeted one of the two closest trees (randomly chosen with probabilities reflecting the ratio of distances; the simpler rule consisting in targeting the closest one was not retained as it led the *basic* forager to be trapped in a periodic series of visits to a limited number of locations). The *sampling*, *associative* or *chronological* forager targeted the tree characterized by the highest ratio between the probability $p_k(t)$ of finding resources at any location known to be occupied by species $k$ (i.e. the probability that a tree of species $k$ provided fruit at the current time $t$) and the travelling distance to reach this location. Such short-term maximization based on a one-step ahead rule is more cognitively realistic than, and almost as efficient in the long term as, multi-step planning [6,31]. The way the probability $p_k(t)$ was assessed depends on the type of memory (see below).

### 2.2.3. Foraging efficiency

We computed the foraging efficiency as the ratio between the total number of fruit trees the forager fed on and the total path length. This ratio corresponds to a long-term efficiency measure which has been classically used as a fitness proxy in the optimal foraging theory (e.g. [32]). A fully naive forager (i.e. without any spatial and temporal memories) but moving along with a straight line to avoid back-tracking would have an efficiency of $4.56 \times 10^{-3}$ exploited tree per unit length (corresponding to the triple product of the tree density, the perception width and the probability for a tree to be productive). The efficiencies of the five types of non-naive foragers (endowed with a perfect spatial memory) were assessed using computer simulations. Because the peak date of any resource type was drawn at random over a year, the production season of some types may straddle two years. Simulations were thus run over two consecutive years and the foraging efficiency was assessed for the second one. We compared the foraging efficiencies of the aforementioned five types of foragers in spatially and temporally homogeneous and heterogeneous environments.

## 2.3. Types of temporal memory

The five types of forager—*basic*, *sampling*, *associative*, *chronological* and *prescient*—all had a perfect long-term spatial memory (i.e. knew the location and species of every fruit tree). They differed only in their skills in processing temporal information. The *prescient* forager knew which trees were productive at any time. The other four types of foragers had to rely on their respective memories to choose the next tree to visit. Contrarily to the *prescient* forager, they were able to determine whether a given tree was productive or empty only if this tree was within their perceptual range.

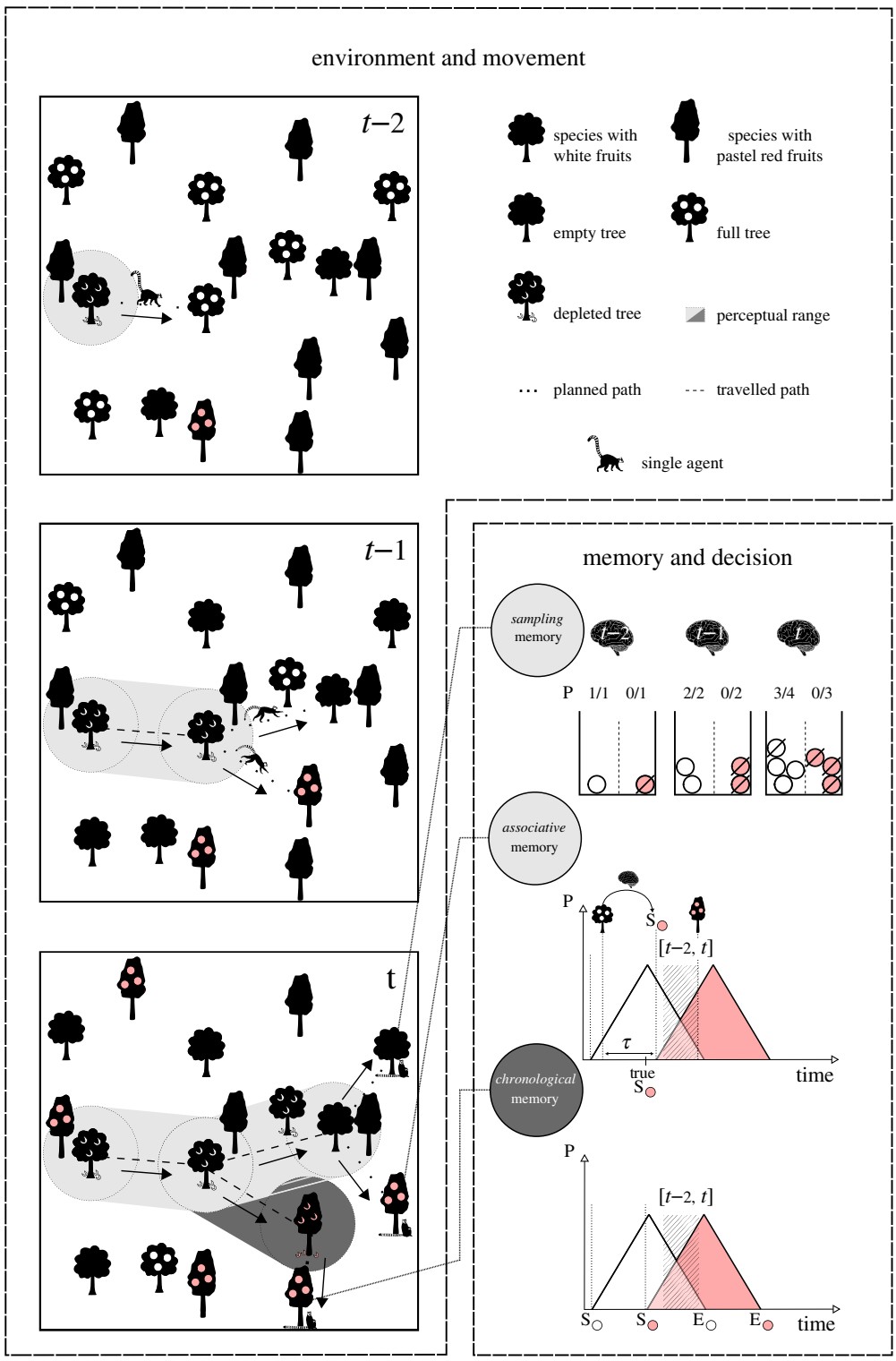

**Figure 2.** Illustration of the working flow of the model. For simplicity, this figure displays an environment with only two types of resource represented by a few trees belonging to two species, and only a few travelling steps are considered. The target tree was the tree with the highest ratio between the probability of yielding fruit (as estimated by the forager depending on the type of memory used) and the distance to reach it. All productive trees encountered en route (i.e. within the perceptual range) were depleted as well at no cost. The *sampling* forager was simply assessing the probability that a given tree yielded fruits based on its recent experience about the proportion of trees of the same species that were productive. The *chronological* forager computed this probability based on its knowledge of the starting (S) and ending (E) date of fruit production period for each species, given the triangular shape of the probability density function of a tree to start yielding fruit (P). The *associative* forager assessed this probability based on its knowledge of the delays between production periods of pairs of species ($\tau$). Thus, when it had detected the start or end of the production period by a species used as a predictor (here the start of production of the white fruit), it could infer the starting and ending dates of production of another species (here the red fruit tree species).

### 2.3.1. Waiting memory

The *basic*, *sampling*, *associative* and *chronological* were all endowed with a *waiting* memory. It is very simple short-term (working) memory which just enabled them to avoid targeting the trees that were visited for the last 15 tu (half of the fruit availability duration of any given tree). The *basic* forager relied only on the *waiting* memory (in addition to the spatial memory). This is why it could not assess the probability $p_k(t)$ that a fruit tree belonging to species $k$ yielded fruit at current time $t$. The other three types of foragers estimated this probability by relying on one of the three specific memories described below.

### 2.3.2. Sampling memory

For each resource type, this short-term (working) and continuously updated memory stored the numbers of productive and empty trees that were recently encountered. The *sampling* forager then estimated the probability $p_k(t)$ as the proportion of productive trees belonging to species $k$ it recently encountered (see electronic supplementary material S1 for details). In the main simulations, *sampling* was based on the food source encountered for the last 15 tu. Additional simulations were performed with other time widths in the range 1 to 60 tu, to investigate to which extent the foraging efficiency was dependent on this time width (see electronic supplementary material S2).

### 2.3.3. Associative memory

For each tree species, this long-term (reference) memory stored the duration of the temporal distribution of fruit production, and the delay between the production start date and a previous event corresponding to either the start or the end of the production of another species (see electronic supplementary material S1 for details). This memory, assumed to result from the experience of tree species phenology over several years, was therefore provided to the *associative* forager before the simulations started and used by this forager during the simulations to assess the probability $p_k(t)$ that fruit was available at any given location $\mathbf{x}_{k,i}$.

### 2.3.4. Chronological memory

For each tree species, this long-term (reference) memory stored the usual start and end dates of fruit production. This memory, assumed to result from experience over several years, was therefore provided the *chronological* forager before the simulations started and was used by this forager during the simulations to directly compute the probability $p_k(t)$ based on the distribution of production of species $k$ (see electronic supplementary material S1 for details). The probability value assessed was obviously accurate for a 'usual' (i.e. average) year, but was likely to be dramatically biased if the current year was characterized by an overall early or late fruit production (see electronic supplementary material S3).

## 3. Results

The five types of foragers differed in their ability to exploit temporal information. Overall, their efficiencies were higher (1.2 to 1.4 times) when the environment was spatially heterogeneous than when it was spatially homogeneous. By contrast, the relationships between foraging efficiencies and the other parameters considered (temporal homogeneity versus heterogeneity, desynchrony level and the type of temporal knowledge) were unaffected by the type of spatial distribution of resources (figure 3).

The foraging efficiencies of the *basic* and the *prescient* foragers were unaffected by the type of temporal distribution of resources (temporal homogeneous versus heterogeneous environment) or by the desynchrony level. Depending on whether the environment was spatially homogeneous or heterogeneous, the spatially omniscient but temporally ignorant *basic* forager was 1.4 times or 1.9 times more efficient than a fully naive forager but 3.0 or 3.2 times less efficient than the *prescient* forager (i.e. who knew at any time where food was available). The other three types of foragers—*sampling*, *associative* and *chronological*—which relied on long-term knowledge about absolute fruit production timing, on long-term knowledge about relative fruit production timing and on short-term knowledge based on recent experience, respectively, had intermediate efficiencies, irrespective of whether the environment was spatially or temporally homogeneous or heterogeneous (figure 3).

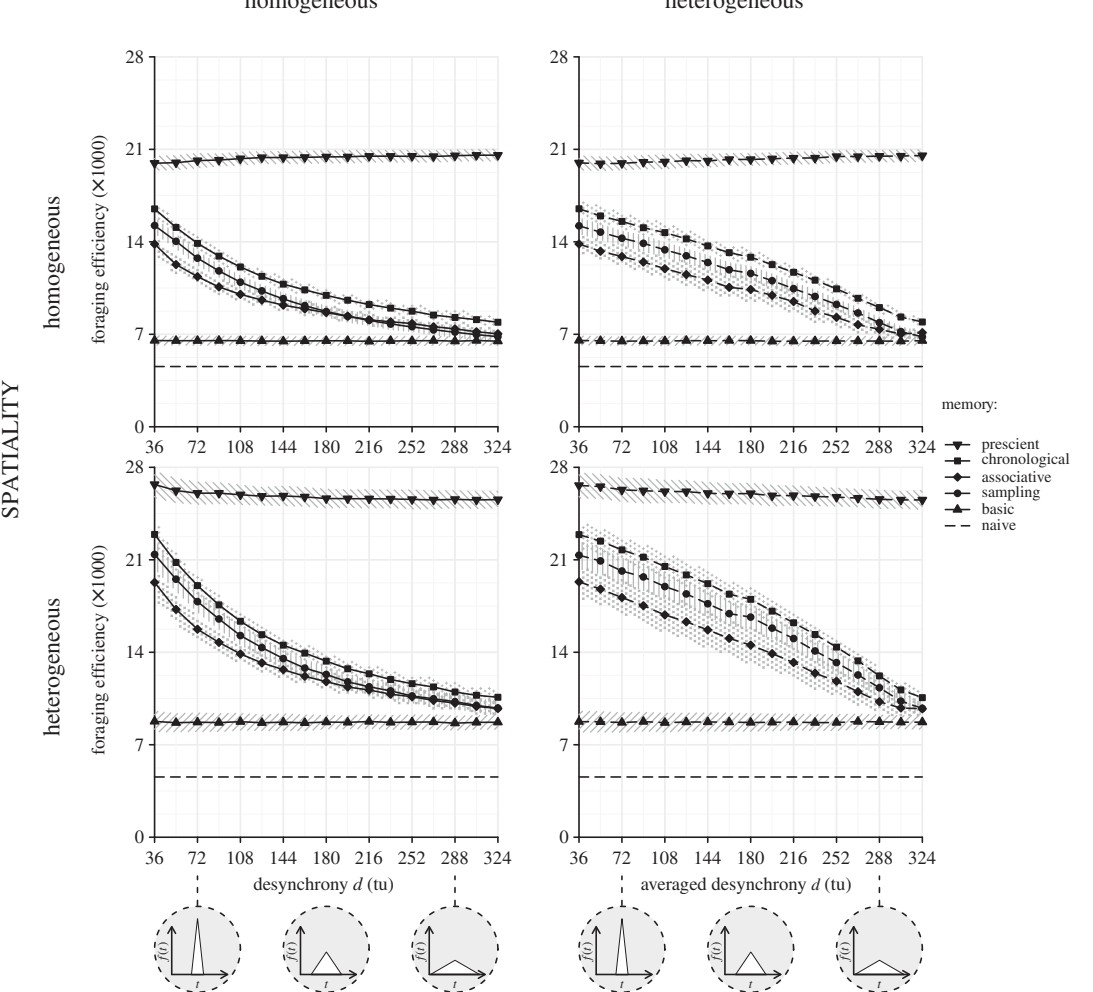

**Figure 3.** Foraging efficiency of the five types of forager as a function of the intra-specific desynchrony of fruit production in temporally and spatially homogeneous or heterogeneous environments. Each point represents the mean foraging efficiency based on 300 simulations, and the pattern-filled backgrounds stand for the corresponding standard deviations. The grey circles illustrate the intra-specific desynchrony level.

As expected, because they were neither temporally naive as the *basic* forager nor temporally omniscient as the *prescient* forager, these three types of foragers were less efficient when the environment became less predictable (i.e. when the desynchrony level was higher; figure 3). The *chronological* forager was always the most efficient of the three, and the *associative* forager was most often the least efficient. To compare the efficiencies of these three types of foragers independently from the arbitrary units used, we rescaled the efficiencies such that the *basic* forager scored at 0 and the *prescient* forager scored at 100 in any type of environment. The score difference between the *chronological* and *sampling* foragers ranged between 4 and 9, while the score difference between the *associative* and *sampling* foragers ranged between –13 and 2 (figure 4). Furthermore, the *sampling* forager could increase its score by 3 or 4 by using a shorter memory span of (5 to 10 tu rather than 15 tu; see electronic supplementary material S2). By contrast, as it relied on knowledge about absolute production timing, the *chronological* forager could easily be impaired by temporal shifts in the resource production (even if a backward shift, i.e. an early season, could be mitigated by a correction mechanism; see electronic supplementary material S3).

Interestingly, the form of the inverse relationship between efficiencies of the *sampling*, *associative* and *chronological* foragers and the desynchrony level differed when the environment was temporally homogeneous (convex function) or heterogeneous (slightly concave function; figure 3). This resulted in efficiencies of the *sampling*, *associative* and *chronological* foragers that were higher in temporally heterogeneous than homogeneous environments. The maximum difference was reached at

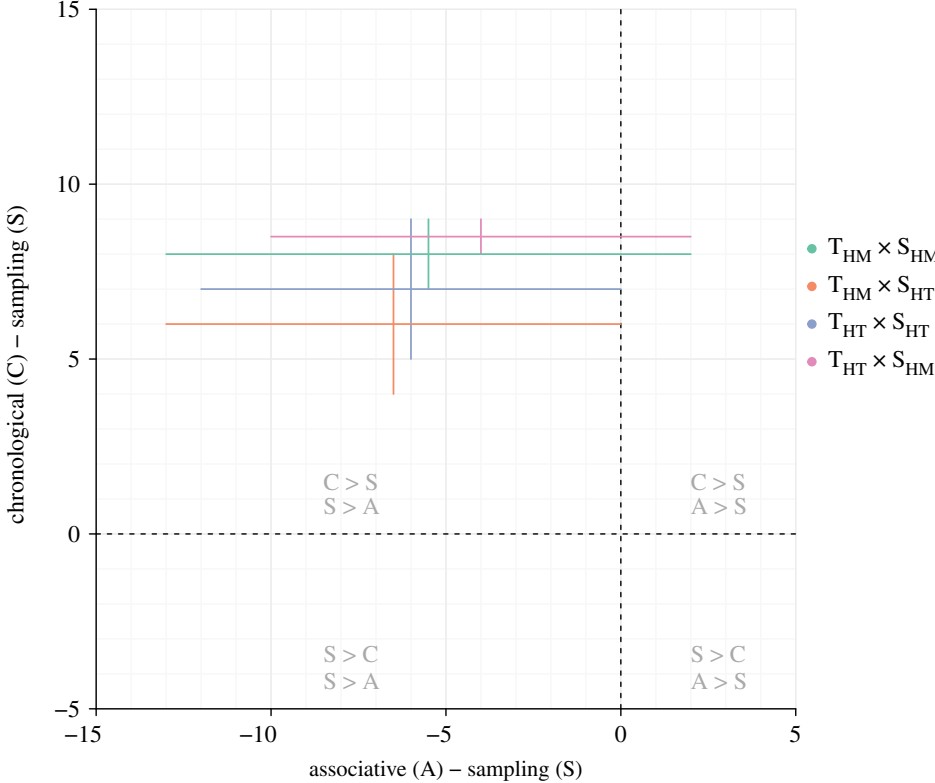

**Figure 4.** Ranges of standardized differences in foraging efficiency between long-term (associative and chronological) and short-term (sampling) temporal memory, in different environmental contexts. $T_{HM}$: temporally homogeneous environment, $T_{HT}$: temporally heterogeneous environment, $S_{HM}$: spatially homogeneous environment, $S_{HT}$: spatially heterogeneous environment.

intermediate desynchrony levels, i.e. at the maximum level of temporal heterogeneity. For a desynchrony level of 180 tu, the score differences between the temporally heterogeneous and homogeneous environments were 21 or 25, 13 or 16 and 21 or 27 for the *sampling*, *associative* and *chronological* foragers, depending on whether the environment was spatially homogeneous or heterogeneous. To sum up, when all tree species were characterized by a high or low desynchrony level, the environment tended to be temporally homogeneous, and the benefits brought by temporal cognition were low or high, respectively. By contrast, for intermediate levels of desynchrony, these benefits appeared higher when the forager fed on tree species characterized by markedly different desynchrony levels.

## 4. Discussion

Our mechanistic model did not aim to mimic any specific species of forager. It was deliberately oversimplified to provide general predictions on the foraging benefits brought by different types of temporal cognition, when resources are available at fixed locations and belonged to different types that renew periodically. When the environment is spatially predictable for an animal with an accurate spatial memory and is also more or less temporally predictable depending on the level of intra-type synchrony of the resources, we show that temporal cognition dramatically improves the foraging efficiency. The *sampling*, *associative* and *chronological* foragers, which can be considered as archetypes of realistic forms of temporal cognition, appeared to be more efficient when the environment was more predictable (i.e. when the within-type desynchrony levels were lower) and, more surprisingly, when it was temporally heterogeneous. Our results thus support the hypothesis of a stronger selection pressure towards cognition for foragers with a relatively wide diet [33], but this hypothesis certainly requires additional results to be tested more directly. A wider diet probably involves remembering more numerous locations and more various attributes of potential food sources. It is also more likely to include resource types with various synchrony levels. Temporal cognition was shown to bring larger benefits to a forager feeding on such resource types.

The exploitation of patchy and ephemeral resources seems to require a high level of information processing [34]. However, the types of memory that are actually used by animals to take advantage of temporal predictability have not yet been clearly empirically identified. The *waiting* memory seems widespread: for example, buffalos [21], elephants [22], capuchins [23] and mangabeys [35] avoid returning to the resource patches they recently exploited. However, a simple *waiting* memory does not enable an animal to benefit from the predictability of seasonally available resources. Accordingly, the foraging efficiency reached by a forager endowed with a *waiting* memory, but without an *associative*, *chronological* or *sampling* memory, remains low in an environment where resources of a given type are not always available.

Evidence that animals can use a *chronological* memory comes from studies on the timing of migration of birds [36] and of monarch butterfly [37]. This behaviour was shown to be triggered by long-term periodic cues such as photoperiod. However, evidence for the use in a foraging context of a *chronological* memory at a large time scale (i.e. beyond the alternation of day and night, which provides obvious periodic cues at short time scales) is scarce (e.g. primates [38]), despite the *chronological* forager outperformed the *associative* and *sampling* foragers in our simulations. This type of memory is probably inefficient for everyday foraging purpose, because of the lack of reliable chronological cues with a sufficiently high temporal resolution, and probably also because its benefits are dramatically reduced for years when the availability of resources is shifted with respect to a 'standard' year (e.g. early or late fruiting). Yet the phenology of numerous resources is highly correlated with season-dependent abiotic factors such as temperature, rainfall or solar radiation, which can lead to synchronizing the production period within and across certain species [39,40], and thereby could be used as cues by *chronological* foragers to target specific types of resource at specific times of year.

*Associative* learning and memory have been evidenced in different contexts [7,41–43]. An *associative* forager is expected to rely on the relatively constant delays between the phenologies of different resource types (e.g. leaves, flower, fruit and/or seed of different species), characterized by marked interspecific synchronies across years and which react more or less quickly to the same synchronizing factors [44]. An *associative* forager may also have learnt that resources produced by some species may become available soon after an episodic meteorological event such heavy rains or a few consecutive days of hot temperature and sun [45]. However, the relatively low efficiency of the *associative* forager in our simulations suggests that a simple sampling memory may usually be sufficient for an animal to forage efficiently.

Environmental sampling is a commonly used process in many animal species [1,2]. For example, primates can infer the resource availability at a not-yet visited patch through the sampling of the environment over the recent past [35,46]. For animals living in a group or colony, sampling information on the spatio-temporal availability of resources can be public (i.e. indirectly provided by the behaviour of other group members [47]). For inferring when the resource of a given type is available, a *sampling* forager is less constrained than a *chronological* or *associative* forager by the timing of resource occurrences: it only requires some level of intra-type resource synchrony (there is no need of periodic renewal). We show here that a simple *sampling* memory enables a forager to reach an efficiency that is similar to the efficiency enabled by a long-term temporal memory. Thus, in an environment that is both spatially predictable (resource patches located at the same place) and temporally more or less predictable (resource types available at specific times), a long-term temporal memory is not necessary to forage efficiently.

If the short-term temporal information stored in a *sampling* memory is sufficient to enable an animal to forage as efficiently as long-term temporal cognition, one can wonder under which selection pressures long-term temporal memories such as the *associative* or *chronological* memories have emerged. We suggest that this is related to the need to anticipate environmental changes. The ability to perform some behaviour in advance with respect to the current needs is likely to exist in numerous animal species [48–50]. It requires long-term temporal knowledge, either learned during the individual lifetime or genetically encoded as found in long-distance migrants [51]. In this case, the ability to anticipate is particularly important because of the long delay that is expected between the time of departure towards a targeted location and the time of arrival. Anticipation may also be useful in a competitive context [52]. Even if the effect of competition can be mitigated by simply avoiding areas that are regularly depleted by competitors [24], prioritizing foraging places to exploit before competitors might be highly advantageous, as observed in baboons [53]. The ability to anticipate cannot be based only on a *sampling* memory, because the information this memory stores is valid only at very short term. Only a long-term temporal memory, which explicitly stores information on (absolute or relative) times of events [54,55], can provide relevant predictions of future events, whereas a short-term temporal

memory, such as the *sampling* memory, is restricted to provide a prediction of current events. We hope our study will encourage future research focusing on temporal memory skills observed in different animal species.

Data accessibility. Data and relevant code for this research work are stored in GitHub (https://github.com/benjaminrobira/Temporal_memory_and_foraging_efficiency) and have been archived within the Zenodo repository (https://doi.org/10.5281/zenodo.5181117)
    The data are provided in the electronic supplementary material [56].
Authors' contribution. B.R. conceived the study with the help of all authors. B.R. coded the model under S.B.'s supervision. B.R. and S.B. wrote the first draft and all authors contributed substantially to improving it. All authors gave final approval for publication and agree to be held accountable for the work performed therein.
Competing interests. The authors declare having no conflict of interest.
Funding. B.R. was funded by a PhD scholarship from the French Ministry of Higher Education and Research to the 'École Normale Supérieure de Paris'. L.R.-L was funded by a Marie Skłodowska-Curie Individual Fellowship from the EU's Horizon 2020 Research and Innovation Programme (grant no. 794760). We also thank the 'Action Transversale' du Muséum National d'Histoire Naturelle (MNHN), the UMR 7206 Eco-Anthropologie of the MNHN and the 'Projet Fédérateur' of the Department 'Homme et Environnement' for their logistical support. We received no funding for this study.
Acknowledgements. We thank two anonymous reviewers for their constructive comments on a previous draft of the paper.

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
