## [Peer Review File · Royal Society Open Science]

Review History

RSOS-210809.R0 (Original submission)

Review form: Reviewer 1

Is the manuscript scientifically sound in its present form?

Yes

Are the interpretations and conclusions justified by the results?

Yes

Is the language acceptable?

Yes

Do you have any ethical concerns with this paper?

No

Have you any concerns about statistical analyses in this paper?

No

Recommendation?

Accept with minor revision (please list in comments)

Comments to the Author(s)

Please see attached file for reviewer comments (Appendix A).

Review form: Reviewer 2**Is the manuscript scientifically sound in its present form?**

Yes

Are the interpretations and conclusions justified by the results?

Yes

Is the language acceptable?

Yes

Do you have any ethical concerns with this paper?

No

Have you any concerns about statistical analyses in this paper?

No

Recommendation?

Accept with minor revision (please list in comments)

Comments to the Author(s)

Note: This is written in markdown format. I've also attached a pdf (see Appendix B) which may be easier to read.

Review for:

Foraging efficiency in temporally predictable environments: Is a long-term temporal memory really advantageous?

This is a much revised manuscript I had previously reviewed. The authors have done a wonderful job incorporating comments from the previous reviewers. The results and figures are clear, and their interpretation of said results in the discussion is sound. Perhaps the only thing I think should be made explicit is some of the modeling assumptions (e.g., that this is a generalist species), but this is relatively minor. Aside from that, I have a few very minor comments that I hope the authors will find useful.

Abstract

Line 24: What is meant by temporal cognitive ability.

Introduction

Line 57: What is mnesically? Looking up that word on a search engine did not return a definition.

Line 86-89: Something is up with this sentence, seems as if a word is not used correctly.

Material and Methods

Line 109: It may be simpler to say "To add environmental context, we assumed food sources are different species of fruit trees, but food sources could correspond to any ephemeral resource." Maybe you could remove "ephemeral" in this suggestion.

Line 111 - 113: This is a great location in the MS to get in front of the general modeling approach the authors took that did not consider a specific species. Yet, a few lines later the authors assume that this species has a perfect spatial memory, which in a way loses this generality (not all species have perfect spatial memory). The species is also an extreme generalist (in that it readily feeds on any fruiting tree species), which is another assumption that is not explicitly made. Perhaps it may be better to flip this logic a little bit and put the focus on the point that is brought up on 113-114. Something like "We made a few assumptions to avoid interferences that may blur the benefits of different temporal memory types." From there the authors could list those types of assumptions and why they are warranted. Regardless, it is very refreshing to see this brought up in the methods rather than much later in the discussion (so a reader does not have to stew on this thought).

Line 127: Why 30 tree species?

Line 143: A sentence that explains what high values of `d_k` vs low values represent would ease interpretation.

Line 158: I may have missed it earlier, but I am not sure what "one of the 17 values evenly distributed..." means. The use of "the" makes me think that this is something that is referenced before, which is not. Perhaps the authors meant that `d_k` took one of 17 evenly distributed values between 36 and 324? If so, please make this a little more clear. This also applies to the 15 values on line 163.

Line 200 - 202: Maybe remove this sentence. This is under the general rules section so a bit about how the general rules differ among foraging types is not quite necessary.

Line 235: What were these specific memories? Maybe it would help to add those memory types in this sentence and add a (described below)?

Line 290 - 299: I wonder if this would be better illustrated graphically, or as a table? Graphically, it could possibly be sub-plots with forager type on x and y axis, and each plot would have 2 points (comparison of row forager vs column forager in the two habitat types). The authors have already made some excellent figures.

Discussion

Line 329: I agree with the authors here, but it would seem as if there would need to be a comparison between a landscape with only a few different resource types to place this statement on a more solid foundation.

Line 333: shown, not showed. Here and throughout other parts of the discussion (though showed is used correctly on line 377).

Line 370: The opening sentence here is a little too general. Sampling of what? If you relate to foraging this will likely become more clear.

Line 378: Maybe it would be better to couch the efficiency here in terms of the other foraging types? There is a lot of uncertainty in what a "high foraging efficiency is here", and all of these are relative to the types of foragers you modeled.

Lines 382 - 401: A great closing paragraph. Not really necessary to add, but have the authors considered how the different foraging types may be favored or not as a result of global climate change (e.g., phenological mismatch would be very bad for chronological foragers).

Figures

The figures are great! The one part I struggle with is the different foragers in the bottom right of figure 2, specifically the sampling memory forager (the other two are much more clear).

Decision letter (RSOS-210809.R0)

Dear Dr Benhamou

On behalf of the Editors, we are pleased to inform you that your Manuscript RSOS-210809 "Foraging efficiency in temporally predictable environments: Is a long-term temporal memory really advantageous?" has been accepted for publication in Royal Society Open Science subject to minor revision in accordance with the referees' reports. Please find the referees' comments along with any feedback from the Editors below my signature.

Please submit your revised manuscript and required files (see below) no later than 7 days from today's (ie 02-Aug-2021) date. Note: the ScholarOne system will 'lock' if submission of the revision is attempted 7 or more days after the deadline. If you do not think you will be able to meet this deadline please contact the editorial office immediately.

on behalf of Dr Kimberley Mathot (Associate Editor) and Kevin Padian (Subject Editor)
openscience@royalsociety.org

Associate Editor Comments to Author (Dr Kimberley Mathot):

Dear Dr. Benhamou,

I have now received reports from two referees with relevant expertise to evaluate your manuscript. I'm pleased to let you know that both referees were very positive about this work, including one referee who had reviewed an earlier version of the study. This referee felt that the revised version had done a thorough job of addressing the comments raised in the initial review.

Both referees have provided very thorough and constructive suggestions for minor revisions that could be implemented to further improve the manuscript. When you submit a revised version, please provide a point-by-point response to these suggestions indicating how they have been addressed in the revision.

Reviewer comments to Author:

Reviewer: 1

Comments to the Author(s)

Please see attached file for reviewer comments: "RSOS_review_20210621.pdf".

Reviewer: 2

Comments to the Author(s)

Note: This is written in markdown format. I've also attached a pdf which may be easier to read: "foraging_review.pdf"

Review for:

Foraging efficiency in temporally predictable environments: Is a long-term temporal memory really advantageous?

This is a much revised manuscript I had previously reviewed. The authors have done a wonderful job incorporating comments from the previous reviewers. The results and figures are clear, and their interpretation of said results in the discussion is sound. Perhaps the only thing I think should be made explicit is some of the modeling assumptions (e.g., that this is a generalist species), but this is relatively minor. Aside from that, I have a few very minor comments that I hope the authors will find useful.

Abstract

Line 24: What is meant by temporal cognitive ability.

Introduction

Line 57: What is mnesically? Looking up that word on a search engine did not return a definition.

Line 86-89: Something is up with this sentence, seems as if a word is not used correctly.

Material and Methods

Line 109: It may be simpler to say "To add environmental context, we assumed food sources are different species of fruit trees, but food sources could correspond to any ephemeral resource." Maybe you could remove "ephemeral" in this suggestion.

Line 111 - 113: This is a great location in the MS to get in front of the general modeling approach the authors took that did not consider a specific species. Yet, a few lines later the authors assume that this species has a perfect spatial memory, which in a way loses this generality (not all species have perfect spatial memory). The species is also an extreme generalist (in that it readily feeds on any fruiting tree species), which is another assumption that is not explicitly made. Perhaps it may be better to flip this logic a little bit and put the focus on the point that is brought up on 113-114. Something like "We made a few assumptions to avoid interferences that may blur the benefits of different temporal memory types." From there the authors could list those types of assumptions and why they are warranted. Regardless, it is very refreshing to see this brought up in the methods rather than much later in the discussion (so a reader does not have to stew on this thought).

Line 127: Why 30 tree species?

Line 143: A sentence that explains what high values of d_k vs low values represent would ease interpretation.

Line 158: I may have missed it earlier, but I am not sure what "one of the 17 values evenly distributed..." means. The use of "the" makes me think that this is something that is referenced before, which is not. Perhaps the authors meant that d_k took one of 17 evenly distributed values between 36 and 324? If so, please make this a little more clear. This also applies to the 15 values on line 163.

Line 200 - 202: Maybe remove this sentence. This is under the general rules section so a bit about how the general rules differ among foraging types is not quite necessary.

Line 235: What were these specific memories? Maybe it would help to add those memory types in this sentence and add a (described below)?

Line 290 - 299: I wonder if this would be better illustrated graphically, or as a table? Graphically, it could possibly be sub-plots with forager type on x and y axis, and each plot would have 2 points (comparison of row forager vs column forager in the two habitat types). The authors have already made some excellent figures.

Discussion

Line 329: I agree with the authors here, but it would seem as if there would need to be a comparison between a landscape with only a few different resource types to place this statement on a more solid foundation.

Line 333: shown, not showed. Here and throughout other parts of the discussion (though showed is used correctly on line 377).

Line 370: The opening sentence here is a little too general. Sampling of what? If you relate to foraging this will likely become more clear.

Line 378: Maybe it would be better to couch the efficiency here in terms of the other foraging types? There is a lot of uncertainty in what a "high foraging efficiency is here", and all of these are relative to the types of foragers you modeled.

Lines 382 - 401: A great closing paragraph. Not really necessary to add, but have the authors considered how the different foraging types may be favored or not as a result of global climate change (e.g., phonological mismatch would be very bad for chronological foragers).

Figures

The figures are great! The one part I struggle with is the different foragers in the bottom right of figure 2, specifically the sampling memory forager (the other two are much more clear).

===PREPARING YOUR MANUSCRIPT===

===PREPARING YOUR REVISION IN SCHOLARONE===

Please ensure that you include a summary of your paper at Step 2 'Type, Title, & Abstract'. This should be no more than 100 words to explain to a non-scientific audience the key findings of your

research. This will be included in a weekly highlights email circulated by the Royal Society press office to national UK, international, and scientific news outlets to promote your work.

Author's Response to Decision Letter for (RSOS-210809.R0)

See Appendix C.

Decision letter (RSOS-210809.R1)

Dear Dr Benhamou

On behalf of the Editors, we are pleased to inform you that your Manuscript RSOS-210809.R1 "Foraging efficiency in temporally predictable environments: Is a long-term temporal memory really advantageous?" has been accepted for publication in Royal Society Open Science subject to minor revision in accordance with the referees' reports. Please find the referees' comments along with any feedback from the Editors below my signature.

Please submit your revised manuscript and required files (see below) no later than 7 days from today's (ie 02-Sep-2021) date. Note: the ScholarOne system will 'lock' if submission of the revision is attempted 7 or more days after the deadline. If you do not think you will be able to meet this deadline please contact the editorial office immediately.

on behalf of Dr Kimberley Mathot (Associate Editor) and Kevin Padian (Subject Editor)
openscience@royalsociety.org

Associate Editor Comments to Author (Dr Kimberley Mathot):

Associate Editor

Comments to the Author:

Thank you for your careful attention to the comments you received in the last round of review. I am satisfied that the revised version addresses all the comments related to the scientific content of the paper and am happy to recommend your article be accepted.

There are a number of typos in this version of the manuscript however, which should be corrected. I have listed these below. Thank you for submitting your work to Royal Society Open Science.

Line 37: "showed" should be "show"
 Line 59: "experimentally showed" should be "shown experimentally"
 Line 158: add "of" to "a value in the range of"
 Line 318: "we showed that", should be "we show that"
 Line 328: "showed" should be "shown"
 Line 325: rephrase to "...additional result to be tested more directly".
 Line 340: "showed" should be "shown"
 Line 372: "showed" should be "show"

Editor comments:

Thanks for your attention to the reviewers' comments. As you'll see there are a few typographical errors to fix. Good luck.

===PREPARING YOUR MANUSCRIPT===

Your revised paper should include the changes requested by the referees and Editors of your manuscript. You should provide two versions of this manuscript and both versions must be provided in an editable format:
 one version identifying all the changes that have been made (for instance, in coloured highlight, in bold text, or tracked changes);
 a 'clean' version of the new manuscript that incorporates the changes made, but does not highlight them. This version will be used for typesetting.

===PREPARING YOUR REVISION IN SCHOLARONE===

Author's Response to Decision Letter for (RSOS-210809.R1)

See Appendix D.

Decision letter (RSOS-210809.R2)

Dear Dr Benhamou,

I am pleased to inform you that your manuscript entitled "Foraging efficiency in temporally predictable environments: Is a long-term temporal memory really advantageous?" is now accepted for publication in Royal Society Open Science.

on behalf of Dr Kimberley Mathot (Associate Editor) and Kevin Padian (Subject Editor)
openscience@royalsociety.org

Appendix A

Review: Foraging efficiency in temporally predictable environments: Is a long-term temporal memory really advantageous?

June 29, 2021

The authors describe the development of a novel individual-based movement model that compares the utility of different types of temporal memory in patchy landscapes. The manuscript details how these three strategies differ in terms of cumulative foraging output. By testing these relationships in different levels of resource heterogeneity, the model is used to hypothesize how different foraging strategies have adapted in animals.

I have found this manuscript, in general, complete and thorough. The theory behind the individual-based movement model is reasonable and connects to existing literature. The measurement of foraging success is clear enough for the results to be presented very simply and elegantly, and the implications of these results are discussed fairly in the Discussion. I would recommend that this manuscript be accepted with minor revisions. I believe that little, if any, re-analysis will be required from my comments and that, in general, it should be straightforward to implement these changes into the manuscript.

1 Comments

1.1 Abstract

L30: Removing the parentheses around “possibly empty” and adding “and” right after the comma may improve the readability of this sentence.

1.2 Introduction

L51: There is a missing right parenthesis after the reference here.

L57: I am not sure what is meant by the word “mnescially” here. This may just be a completely usable word that I am not familiar with, but I figured I would bring it to the authors’ attention in case it is a typo.

L73: Changing “If” to “While” or something similar at the beginning of this sentence reads better to me - feel free to disregard this suggestion though.

L87: This line reads a bit confusingly; I might recommend changing “...take advantage that...” to “take advantage of the fact that...” or something similar.

1.3 Methods

L109: While I understand why “fruit trees” are chosen as the patch type here, I feel that this explanation can be rewritten. It may be better to lead by highlighting the generality of the proposed model and explaining afterwards that for clarity, patches will be referred to as fruit trees.

L160: Consider changing “...part of the tree species...” to “...some of the tree species...” or something similar.

L184: In this context, does a “temporary resource shortage” imply that every tree in the landscape is devoid of fruit? In that case, I would not expect that to be grounds for the animal to stop moving. For example, a forager may perceive a tree to have fruit ($p_k(t) > 0$) even if it does not.

L244: A reference to Figure S1 here may clarify exactly what was done with regards to this analysis.

1.4 Results

L295: When you mention that the associative forager registered relative scores below 0 (e.g., -12), does this imply that they performed worse than the basic forager? Would this constitute a contradiction with L281, which states that the three informed foragers were always better than the basic forager?

1.5 Supplementary Material

The Word document attached did not have line numbers, but I have added my own. Hopefully this does not add additional confusion.

SL7: I am still a bit confused on why the base length is reduced in the calculation of $p_k(t)$. I may just need clarification personally - but providing some quantitative evidence that “The distribution experienced (over years) by a forager had, on average, the mean value M_k but a reduced base length equal to $b_k \approx 0.875d_k$ ” could be helpful.

SL20: Is the “waiting forager” the same as the “basic forager”?

SL38: Once again, does the associative forager know what b_i is? This would seem to require it to know what d_i is, which may require multiple years of foraging experience. Is this realistic?

SL53: Consider rewording the phrase “A too short sampling span...” to “A sampling span that is too short...”. The same can be said for “... a too long span...” on SL54.

Appendix B

Review for:

Foraging efficiency in temporally predictable environments: Is a long-term temporal memory really advantageous?

This is a much revised manuscript I had previously reviewed. The authors have done a wonderful job incorporating comments from the previous reviewers. The results and figures are clear, and their interpretation of said results in the discussion is sound. Perhaps the only thing I think should be made explicit is some of the modeling assumptions (e.g., that this is a generalist species), but this is relatively minor. Aside from that, I have a few very minor comments that I hope the authors will find useful.

Abstract

Line 24: What is meant by temporal cognitive ability.

Introduction

Line 57: What is mnesically? Looking up that word on a search engine did not return a definition.

Line 86-89: Something is up with this sentence, seems as if a word is not used correctly.

Material and Methods

Line 109: It may be simpler to say "To add environmental context, we assumed food sources are different species of fruit trees, but food sources could correspond to any ephemeral resource." Maybe you could remove "ephemeral" in this suggestion.

Line 111 - 113: This is a great location in the MS to get in front of the general modeling approach the authors took that did not consider a specific species. Yet, a few lines later the authors assume that this species has a perfect spatial memory, which in a way loses this generality (not all species have perfect spatial memory). The species is also an extreme generalist (in that it readily feeds on any fruiting tree species), which is another assumption that is not explicitly made. Perhaps it may be better to flip this logic a little bit and put the focus on the point that is brought up on 113-114. Something like "We made a few assumptions to avoid interferences that may blur the benefits of different temporal memory types." From there the authors could list those types of assumptions and why they are warranted. Regardless, it is very refreshing to see this brought up in the methods rather than much later in the discussion (so a reader does not have to stew on this thought).

Line 127: Why 30 tree species?

Line 143: A sentence that explains what high values of `d_k` vs low values represent would ease interpretation.

Line 158: I may have missed it earlier, but I am not sure what "one of the 17 values evenly distributed..." means. The use of "the" makes me think that this is something that is referenced before, which is not. Perhaps the authors meant that `d_k` took one of 17 evenly distributed values between 36 and 324? If so, please make this a little more clear. This also applies to the 15 values on line 163.

Line 200 - 202: Maybe remove this sentence. This is under the general rules section so a bit about how the general rules differ among foraging types is not quite necessary.

Line 235: What were these specific memories? Maybe it would help to add those memory types in this sentence and add a (described below)?

Line 290 - 299: I wonder if this would be better illustrated graphically, or as a table? Graphically, it could possibly be sub-plots with forager type on x and y axis, and each plot would have 2 points (comparison of row forager vs column forager in the two habitat types). The authors have already made some excellent figures.

Discussion

Line 329: I agree with the authors here, but it would seem as if there would need to be a comparison between a landscape with only a few different resource types to place this statement on a more solid foundation.

Line 333: shown, not showed. Here and throughout other parts of the discussion (though showed is used correctly on line 377).

Line 370: The opening sentence here is a little too general. Sampling of what? If you relate to foraging this will likely become more clear.

Line 378: Maybe it would be better to couch the efficiency here in terms of the other foraging types? There is a lot of uncertainty in what a "high foraging efficiency is here", and all of these are relative to the types of foragers you modeled.

Lines 382 - 401: A great closing paragraph. Not really necessary to add, but have the authors considered how the different foraging types may be favored or not as a result of global climate change (e.g., phenological mismatch would be very bad for chronological foragers).

Figures

The figures are great! The one part I struggle with is the different foragers in the bottom right of figure 2, specifically the sampling memory forager (the other two are much more clear).

Appendix C

Dear Editor,

We are glad to see that our work is suitable for publication in *Royal Society Open Science*. We warmly thank the two reviewers for their positive summaries, the constructive comments they made, and the corrections and suggestions they offered. We have followed most of them. Please find below detailed point-by-point answers.

Sincerely yours,

Simon Benhamou and Benjamin Robira, on behalf of all co-authors.

REVIEWER 1

The authors describe the development of a novel individual-based movement model that compares the utility of different types of temporal memory in patchy landscapes. The manuscript details how these three strategies differ in terms of cumulative foraging output. By testing these relationships in different levels of resource heterogeneity, the model is used to hypothesize how different foraging strategies have adapted in animals. I have found this manuscript, in general, complete and thorough. The theory behind the individual-based movement model is reasonable and connects to existing literature. The measurement of foraging success is clear enough for the results to be presented very simply and elegantly, and the implications of these results are discussed fairly in the Discussion. I would recommend that this manuscript be accepted with minor revisions. I believe that little, if any, re-analysis will be required from my comments and that, in general, it should be straightforward to implement these changes into the manuscript

1 Comments

1.1 Abstract

L30: Removing the parentheses around "possibly empty" and adding "and" right after the comma may improve the readability of this sentence.

Done

1.2 Introduction

L51: There is a missing right parenthesis after the reference here.

Indeed. This has been corrected.

L57: I am not sure what is meant by the word "mnesically" here. This may just be a completely usable word that I am not familiar with, but I figured I would bring it to the authors' attention in case it is a typo.

We replaced the word “mnesically” with “... in memory”.

L73: Changing "If" to "While" or something similar at the beginning of this sentence reads better to me - feel free to disregard this suggestion though.

Done.

L87: This line reads a bit confusingly; I might recommend changing "...take advantage that..." to "take advantage of the fact that..." or something similar.

Done, thank you.

1.3 Methods

L109: While I understand why "fruit trees" are chosen as the patch type here, I feel that this explanation can be rewritten. It may be better to lead by highlighting the generality of the proposed model and explaining afterwards that for clarity, patches will be referred to as fruit trees.

Done.

L160: Consider changing "...part of the tree species..." to "...some of the tree species..." or something similar.

Done.

L184: In this context, does a "temporary resource shortage" imply that every tree in the landscape is devoid of fruit? In that case, I would not expect that to be grounds for the animal to stop moving. For example, a forager may perceive a tree to have fruit ($p_k(t) > 0$) even if it does not.

Depending of the forager species, entering in torpor in case of resource shortage may or may not be realistic. With our model, however, we did not attempt to be realistic but to provide a

fair comparison between the foraging efficiency reached by different types of temporal memory. A very efficient forager is more liable to face resource shortage (due in part to its activity). Consequently, letting the foragers be active when the environment was empty would have result in a decreased contrast between the different types of memory.

L244: A reference to Figure S1 here may clarify exactly what was done with regards to this analysis.

Done.

1.4 Results

L295: When you mention that the associative forager registered relative scores below 0 (e.g., -12), does this imply that they performed worse than the basic forager? Would this constitute a contradiction with L281, which states that the three informed foragers were always better than the basic forager?

The score of the basic and prescient foragers were set to 0 and 100 to provide a convenient scale. The values mentioned afterwards do not correspond to scores but to score **differences** between the other three types of forager. Thus the value "-13" when comparing the associative and sampling foragers means that the score of the associative forager was 13 below the score of the sampling forager, although both have positive scores (i.e. above the score of the basic forager). In the revised version, we simplified the text that was a bit cumbersome and illustrated these results with a new Figure (Fig. 4).

1.5 Supplementary Material

The Word document attached did not have line numbers, but I have added my own. Hopefully this does not add additional confusion.

SL7: I am still a bit confused on why the base length is reduced in the calculation of $pk(t)$. I may just need clarification personally - but providing some quantitative evidence that "The distribution experienced (over years) by a forager had, on average, the mean value M_k but a reduced base length equal to $b_k - 0.875dk$ " could be helpful.

This result was obtained using computer simulations, in measuring repetitively the base length from limited samples of individuals. We specified it in the revised version.

SL20: Is the "waiting forager" the same as the "basic forager"?

Right, it was a typo, thank you.

SL38: Once again, does the associative forager know what b_k is? This would seem to require it to know what d_i is, which may require multiple years of foraging experience. Is this realistic?

Both the chronological and associative foragers were assumed to know what b_k is (the duration of the experienced fruiting period of tree species k). Depending on the lifetime of the forager species considered, this may be realistic or not. As previously mentioned, our aim was not to design a realistic model but to provide a fair comparison between short-term memory (sampling) based foragers which, in particular, do not know the b_k value, and long-term memory (chronological and associative)-based foragers which have long-term (over years) information about the tree species on which they feed.

SL53: Consider rewording the phrase "A too short sampling span..." to "A sampling span that is too short...". The same can be said for "... a too long span..." on SL54.

Done.

REVIEWER 2

This is a much revised manuscript I had previously reviewed. The authors have done a wonderful job incorporating comments from the previous reviewers. The results and figures are clear, and their interpretation of said results in the discussion is sound. Perhaps the only thing I think should be made explicit is some of the modeling assumptions (e.g., that this is a generalist species), but this is relatively minor. Aside from that, I have a few very minor comments that I hope the authors will find useful.

Abstract

Line 24: What is meant by temporal cognitive ability.

To be clearer, we replaced this expression by "abilities to process temporal information".

Introduction

Line 57: What is mnesically? Looking up that word on a search engine did not return a definition.

We replaced "...mnesically associate two events" by "...associate two events in memory".

Line 86-89: Something is up with this sentence, seems as if a word is not used correctly.

Indeed, a few words ("of the fact") were missing between "advantage" and "that"

Material and Methods

Line 109: It may be simpler to say "To add environmental context, we assumed food sources are different species of fruit trees, but food sources could correspond to any ephemeral resource." Maybe you could remove "ephemeral" in this suggestion.

Thank you for the suggestion. We followed it, using a slightly modified version.

Line 111 - 113: This is a great location in the MS to get in front of the general modeling approach the authors took that did not consider a specific species. Yet, a few lines later the authors assume that this species has a perfect spatial memory, which in a way loses this generality (not all species have perfect spatial memory). The species is also an extreme generalist (in that it readily feeds on any fruiting tree species), which is another assumption that is not explicitly made. Perhaps it may be better to flip this logic a little bit and put the focus on the point that is brought up on 113-114. Something like "We made a few assumptions to avoid interferences that may blur the benefits of different temporal memory types." From there the authors could list those types of assumptions and why they are warranted. Regardless, it is very refreshing to see this brought up in the methods rather than much later in the discussion (so a reader does not have to stew on this thought).

We agree that the first sentence of this part could be misinterpreted and we removed it. It should now be clear that our model specifically focuses on temporal information processing, and that other aspects were deliberately oversimplified.

Line 127: Why 30 tree species?

This number is simply a trade-off between the total number of trees that can be computationally managed, the minimum number of trees per species to consider the effect of intra-species synchrony, and the minimum number of species that is required to consider the effect of inter-species synchrony.

Line 143: A sentence that explains what high values of d_k vs low values represent would ease interpretation.

We added such a sentence.

Line 158: I may have missed it earlier, but I am not sure what "one of the 17 values evenly distributed..." means. The use of "the" makes me think that this is something that is referenced before, which is not. Perhaps the authors meant that d_k took one of 17 evenly distributed values between 36 and 324? If so, please make this a little more clear. This also applies to the 15 values on line 163.

We agree that our writing may be misleading. Thus, we rephrased this part more explicitly.

Line 200 - 202: Maybe remove this sentence. This is under the general rules section so a bit about how the general rules differ among foraging types is not quite necessary.

We agree that this sentence is not related to general rules, but we also think that it is important to say right now that the way the probability $p_k(t)$ mentioned above is computed will be explained later, to prevent readers from thinking they missed an explanation. Consequently, we did not remove the sentence but shortened it, mentioning that the explanation will come just later on.

Line 235: What were these specific memories? Maybe it would help to add those memory types in this sentence and add a (described below)?

We agree and changed the sentence accordingly.

Line 290 - 299: I wonder if this would be better illustrated graphically, or as a table? Graphically, it could possibly be sub-plots with forager type on x and y axis, and each plot would have 2 points (comparison of row forager vs column forager in the two habitat types). The authors have already made some excellent figures.

Good suggestion. We now provide a new figure (Fig. 4) to illustrate these numbers.

Discussion

Line 329: I agree with the authors here, but it would seem as if there would need to be a comparison between a landscape with only a few different resource types to place this statement on a more solid foundation.

Indeed, to fully explore this statement, the model should have included more than “two categories” and should also consider the evolution of diet selectivity (which may drive or be driven by memory). This was beyond the scope of our study but stands as an interesting avenue. Accordingly we added at the end of our statement that "this hypothesis certainly requires additional results to be confirmed."

Line 333: shown, not showed. Here and throughout other parts of the discussion (though showed is used correctly on line 377)

It seems that "showed" can be used in place of "shown" for the past participle. However, in our text, we mixed both inconsistently. In the revised version, the used "showed" everywhere.

Line 370: The opening sentence here is a little too general. Sampling of what? If you relate to foraging this will likely become more clear.

We spoke about the sampling process in general here, not necessarily in a foraging context. We added some precisions to make it clearer and think that the examples given just after should enable the reader to well understand what we meant.

Line 378: Maybe it would be better to couch the efficiency here in terms of the other foraging types? There is a lot of uncertainty in what a "high foraging efficiency is here", and all of these are relative to the types of foragers you modeled.

We agree and replaced "high efficiency" by "efficiency that is similar to the efficiency enabled by a long-term temporal memory".

Lines 382 - 401: A great closing paragraph. Not really necessary to add, but have the authors considered how the different foraging types may be favored or not as a result of global climate change (e.g., phenological mismatch would be very bad for chronological foragers).

Thanks for the appreciation. We did not consider climate changes because it would be too speculative to infer their effect about the vegetation phenology across years. It seems to us that early and late seasons may be considered (as we did in the Supplementary Material) without specifically mentioning climate changes. In addition, climate changes may induce phenological mismatch, but might as well considerably simplify the production pattern, as observed in tropical forest where the productivity and synchrony of fruit trees increase (Chapman et al. 2005. A 12-year phenological record of fruiting: implications for frugivore

populations and indicators of climate change. Pp. 75-92 in *Tropical fruits and frugivores*, Springer, Dordrecht).

Figures

The figures are great! The one part I struggle with is the different foragers in the bottom right of figure 2, specifically the sampling memory forager (the other two are much more clear).

We added a forager to the final destination tree to improve the readability of this part of Fig 2.

Appendix D

Dear Editor,

We corrected the typos. Thanks to have point them.

Sincerely yours,

Simon Benhamou, on behalf of all co-authors.